# Genetic map of regional sulcal morphology in the human brain from UK biobank data

Benjamin B. Sun [1,2] ✉, Stephanie J. Loomis[1,10], Fabrizio Pizzagalli[3,4,10], Natalia Shatokhina[4], Jodie N. Painter [5], Christopher N. Foley [6,7], Biogen Biobank Team*, Megan E. Jensen[8], Donald G. McLaren[8], Sai Spandana Chintapalli[9], Alyssa H. Zhu[4], Daniel Dixon[4], Tasfiya Islam[4], Iyad Ba Gari [4], Heiko Runz [1], Sarah E. Medland [5], Paul M. Thompson[4,11] ✉, Neda Jahanshad[4,11] ✉ & Christopher D. Whelan [1,11] ✉

Genetic associations with macroscopic brain structure can provide insights into brain function and disease. However, specific associations with measures of local brain folding are largely under-explored. Here, we conducted large-scale genome- and exome-wide associations of regional cortical sulcal measures derived from magnetic resonance imaging scans of 40,169 individuals in UK Biobank. We discovered 388 regional brain folding associations across 77 genetic loci, with genes in associated loci enriched for expression in the cerebral cortex, neuronal development processes, and differential regulation during early brain development. We integrated brain eQTLs to refine genes for various loci, implicated several genes involved in neurodevelopmental disorders, and highlighted global genetic correlations with neuropsychiatric phenotypes. We provide an interactive 3D visualisation of our summary associations, emphasising added resolution of regional analyses. Our results offer new insights into the genetic architecture of brain folding and provide a resource for future studies of sulcal morphology in health and disease.

Human brain structure and function are complex drivers of basic and higher cognitive processes, which vary between individuals and in numerous neurological, psychiatric and cognitive disorders. Structural magnetic resonance imaging (MRI) scans provide a reliable, non-invasive measure of brain structure and are widely used in research and clinical settings. Genetic variants influencing brain structure and function are important to identify, as they can help uncover pathophysiological pathways involved in heritable brain diseases and prioritise novel targets for drug development. Several large-scale genome-wide association studies (GWAS) have identified genetic influences on variations in brain structure and function[1-3] revealing novel insights into processes guiding brain development, and highlighting potential shared genetic aetiologies with neurodegenerative and psychiatric conditions[4,5].

To date, most neuroimaging GWAS have focused on broad, macroscale anatomical features such as subcortical volume, cortical thickness and white matter microstructure[6]. Anomalies of cortical gyrification - the folding of the cerebral cortex into its characteristic

[1]Translational Biology, Research & Development, Biogen Inc., Cambridge, MA, US. [2]BHF Cardiovascular Epidemiology Unit, Department of Public Health and Primary Care, University of Cambridge, Cambridge, UK. [3]Department of Neuroscience "Rita Levi Montalcini", University of Turin, Turin, Italy. [4]Imaging Genetics Center, Mark and Mary Stevens Neuroimaging and Informatics Institute, Keck School of Medicine, University of Southern California, Marina del Rey, CA, US. [5]QIMR Berghofer Medical Research Institute, Brisbane, QLD, Australia. [6]MRC Biostatistics Unit, School of Clinical Medicine, University of Cambridge, Cambridge, UK. [7]Optima Partners, Edinburgh, UK. [8]Clinical Sciences, Research & Development, Biogen Inc., Cambridge, MA, US. [9]Department of Bioengineering, University of Pennsylvania, Philadelphia, PA, US. [10]These authors contributed equally: Stephanie J. Loomis, Fabrizio Pizzagalli. [11]These authors jointly supervised this work: Paul M. Thompson, Neda Jahanshad, Christopher D. Whelan. *A list of authors and their affiliations appears at the end of the paper. ✉e-mail: bbsun92@outlook.com; pthomp@usc.edu; njahansh@usc.edu; christopherdwhelan@outlook.com

concave sulci (fissures) and convex gyri (ridges) - contribute to many neurodevelopmental and neuropsychiatric conditions[7,8], but the genetic underpinnings of gyrification remain relatively understudied[9]. Sulcal characteristics and folding patterns are altered across a range of neurodevelopmental disorders, from cortical dysplasias[10] to neurogenetic syndromes[11], and radiologists often use sulcal widening as an early indicator of atrophy in degenerative diseases[12], as it offers a clear and sensitive biomarker of disease progression[13,14]. Recent neuroimaging genetics investigations have broadened in scale and scope, examining sulcal morphology across the full brain[15,16], but primarily focusing on isolated sulcal descriptors such as depth[15], and largely overlooking localised effects.

Using four independent datasets, we recently outlined a range of heritable sulcal measures that can be reliably quantified at high resolution across the whole brain, irrespective of MRI platform or acquisition parameters[17]: sulcal depth, length, width and surface area. Sulcal depth represents the distance between the cortical surface and the exposed, gyral surface (also known as the hull[18]). Sulcal length represents the distance of the intersection between the medial sulcal surface and the hull. Sulcal width, also known as sulcal span[19] or fold opening[20], represents the distance between each gyral bank, averaged over all points spanning the median sulcal surface[18]. Sulcal surface area represents a composite of sulcal width, depth and length measures. Sulcal descriptors strongly correlate with measures of thickness in their adjacent cortical regions[21]; however, sulcal measures are likely more sensitive to increased age[21], cognitive performance[22], and genetic effects[17] compared with more commonly analysed metrics of gyral morphometry.

Here we conducted a comprehensive genome-wide analysis of four regional sulcal shape parameters, extracted from the multi-centre brain MRI scans of 40,169 participants in the UK Biobank. To discover rare and common genetic variants influencing cortical gyrification, we conducted GWAS and exome-wide analysis of a total of 450 sulcal parameters[17]. Sulcal shape descriptors, comprising length, mean depth, width, and surface area, were extracted from a discovery cohort of 26,530 individuals of European ancestry and a replication cohort of 13,639 individuals. After mapping the genetic architecture of regional sulcal measures across the cortex, we highlight putative biological and developmental pathway involvement as well as links to neuropsychiatric conditions. Finally, we provide a portal to interactively visualise our results in 3D (https://enigma-brain.org/sulci-browser), demonstrating various complex patterns of associations, to help inform future investigations of human cortical morphology.

## Results

Regional brain sulcal measurements (including sulcal length, width, mean depth and surface area), regional delineations, and phenotype nomenclature are summarised in Supplementary Data 1 and Fig. 1a. Reliability of the method used to extract sulcal characteristics has been extensively outlined in[17] and summarised in Supplementary Information. We determined the overall clustering of the high-dimensional sulcal phenotypes through nonlinear dimensionality reduction using t-distributed stochastic neighbour embedding (t-SNE)[23], revealing that sulcal measures formed distinct clusters compared with existing UK Biobank brain imaging phenotypes (Supplementary Fig. 1a, Supplementary Data 2, Methods). In addition, we found that sulcal width parameters formed a distinct cluster compared to the other three sulcal shape parameters (Supplementary Fig. 1b). Notably, for sulcal width in particular, the t-SNE representation broadly retained expected brain lobe organisation (Fig. 1b, bottom left). Phenotypes with missingness >75% were excluded from subsequent analysis, leaving 450 measurements (224 left and 225 right hemisphere measures) for analysis.

### Genetic architecture of regional brain sulcal folds

We conducted GWAS of 450 independent regional brain sulcal measurements across the left and right hemispheres for 11.9 million combined imputed and whole-exome sequenced variants in UKB participants, divided into a discovery cohort ($n = 26,530$) and a replication cohort ($n = 13,639$) (Methods, Supplementary Fig. 2).

At a significance threshold of $p < 2 \times 10^{-10}$, which accounts for the effective number of independent sulcal measures analysed (Methods), we found and replicated - at $p < 0.05$ - a total of 186 specific sulcal parameter associations (for at least one hemisphere) across 41 genetic loci (388 associations across 77 loci at $p < 5 \times 10^{-8}$) (Fig. 1c, Supplementary Data 3). We also performed GWAS on 220 additional bilateral sulcal measurements, averaging values from left and right brain hemispheres, identifying a total of 162 replicated associations across 47 loci at $p < 2 \times 10^{-10}$ (335 associations across 107 loci at $p < 5 \times 10^{-8}$), where 6 (across 3 loci) and 108 additional associations (across 42 loci) were also found at $p < 2 \times 10^{-10}$ and $p < 5 \times 10^{-8}$ respectively (Supplementary Data 4). We performed association testing using permuted samples ($n = 100$ times) to empirically estimate the expected number of associations that also replicate under the null (Supplementary Information). Even at nominal GWAS significance ($p < 5 \times 10^{-8}$) and replication ($p < 0.05$) thresholds, the median false discovery rate was -1% (99th percentile = 2.1%), suggesting false positive discoveries were well-calibrated below 5% (Supplementary Information). Genomic inflation was well controlled (median λgc = 1.02, range: 0.99–1.07). We found an inverse relationship between effect sizes and minor allele frequency (MAF) (Supplementary Fig. 3), in line with other disease and intermediate trait results, and consistent with the assumption that variants showing strong effects are deleterious and rarer. Additional sensitivity analyses, adjusting for (1) effects of single X/Y/Z plane head scaling and (2) potential effects of cortical thickness and surface area, revealed that sulcal associations are largely independent of these effects (Supplementary Information).

We found a similar number of statistically significant associations for left- and right-hemispheric traits. Approximately 59% of significant associations were with sulcal width, followed by 17% of associations with mean sulcal depth, 16% with sulcal surface area, and 8% with sulcal length measures (Supplementary Data 5). These genetic association patterns are consistent with their heritability estimates and with prior studies[17] indicating that sulcal width represents the most heritable measure, followed by sulcal depth, surface area and length (Supplementary Fig. 4). Comparing the absolute Z-scores of lead associations across hemispheres (left, right and bilateral, Supplementary Fig. 5), we found no significant difference between left and right hemisphere (paired t-test $p = 0.25$). Bilateral associations tended to exhibit stronger associations (mean abs(Z-score) 1.00 higher *vs* right, $p = 4.1 \times 10^{-96}$ and 0.92 higher *vs* left, $p = 1.8 \times 10^{-82}$), consistent with their heritability estimates (Supplementary Figs. 4 and 5).

Some genetic loci exhibited highly pleiotropic associations across multiple brain regions; for example, 10 genetic loci were associated with 10 or more sulcal measures, showing different association patterns across sulcal parameters. Notably, the chr1:215 Mb locus (near *KCNK2*) and chr12:106 Mb locus (12q23.3, *NUAK1*) were associated with 23 and 22 width measures respectively across multiple brain regions. Similarly, the chr16:87 Mb locus (6q24.2, near *C16orf95*) was associated with 16 width measures across multiple brain regions, 4 mean depth measures and 1 surface area measure, mostly in the frontal lobe. The chr17:47 Mb locus (17q21.31, containing *MAPT* and *KANSL1*) was associated with 16 width, 9 surface area, 6 mean depth and 2 length measures, mostly in the temporal and calcarine-occipital regions, whilst the chr6:126 Mb locus (6q22.32, containing *CENPW*) was associated with 9 surface area, 4 length, 4 mean depth and 2 width measures - mostly in the frontal and calcarine-occipital regions (Fig. 1c, Supplementary Data 3). We performed multi-trait colocalization and clustering using HyPrColoc[24] across loci associated with ≥2 sulcal measures ($n = 72$ loci, Supplementary Data 6) and found 62 (86%) of the loci colocalised to a single cluster, 8 (11%) to two clusters, and one to three clusters. The *MAPT-KANSL1* region contained a large number of sulcal

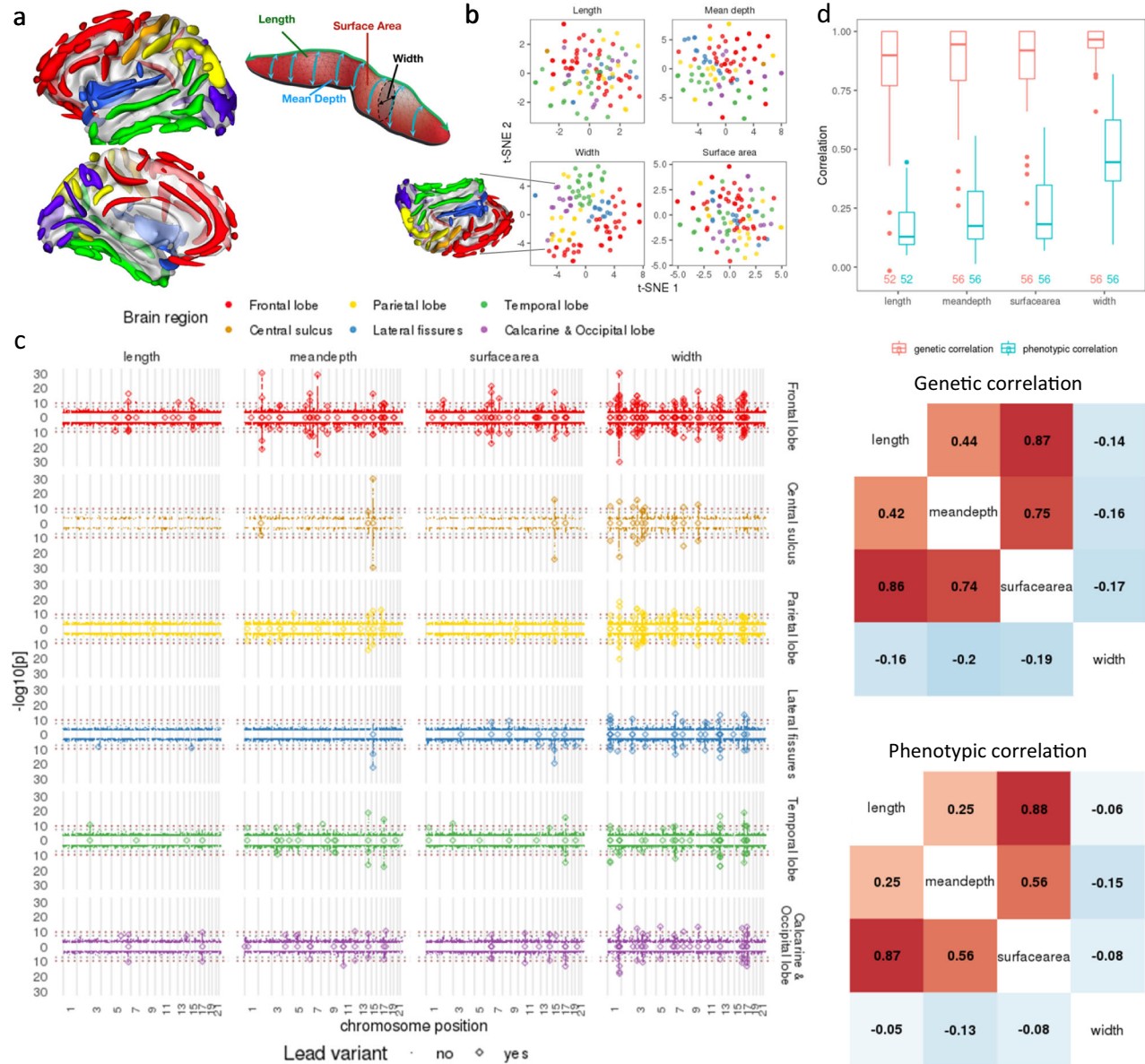

**Fig. 1 | Summary of brain sulcal association results. a** Schematic of brain sulcal folds and shape parameters. Brain region legend corresponds to colours in figures **a**–**c**. **b** t-SNE of regional brain sulcal measures for each shape parameter. **c** Manhattan plots by brain region, shape parameter and side. Diamonds indicate lead associations that replicated ($p < 0.05$). $N = 26,530$ (discovery) and $n = 13,639$ (replication) biologically independent sample measures. Points above 0 in the y-axis in each plot refers to associations with left sided sulcal measures, below 0 with right sided measures. Diamonds along 0 in the y-axis indicate lead associations for bilateral sulcal measures. Dashed horizonal line indicate GWAS significance thresholds (grey: $p = 5 \times 10^{-8}$, dark red: $p = 2 \times 10^{-10}$). High resolution of the figure is

available in the Supplementary Files. **d** Top: boxplot of genetic and phenotypic correlations between left and right sides. Number of sulcal traits listed beneath the boxplot. Each box plot presents the median, first and third quartiles, with upper and lower whiskers representing 1.5× inter-quartile range above and below the third and first quartiles respectively. Middle: Genetic correlation between shape parameters. Bottom: Phenotypic correlation between shape parameters. Middle and bottom: left hemisphere correlations in upper triangle, right hemisphere correlations in lower triangle. Phenotypic and genetic correlations were averaged across each hemisphere separately per sulcal parameter. Extended phenotypic and genetic correlation heatmap is shown in Supplementary Fig. 6.

associations which were not colocalised with others in the region (Supplementary Data 6).

We cross-referenced the replicated lead variants and their proxies ($r^2 > 0.8$) for significant ($p < 5 \times 10^{-8}$) associations in prior brain imaging GWAS studies of non-sulcal traits[6] (LD proxy $r^2 > 0.8$, +/−500 Kb around the lead variant; see Supplementary Information). We found that 56 of the 119 loci identified at $p < 5 \times 10^{-8}$, and 23 of the 44 loci identified at $p < 2 \times 10^{-10}$, were associated with non-sulcal brain imaging phenotypes of brain volume, surface area and white matter microstructure. The 10 highly pleiotropic genetic loci identified in this study (e.g. *CENPW*-containing locus 6q22.32, *MAPT-KANSL1*-containing locus

17q21.31, *C16orf95* locus 6q24.2, *NUAK1* locus 12q23.3) have been implicated across multiple prior neuroimaging studies (Supplementary Data 7). Approximately half of all replicated associations were not previously implicated in published genetic studies of non-sulcal brain imaging endpoints; this may, however, reflect differences in the parcellation strategies or the sample sizes achieved across studies rather than any unique genetic influences on sulcal parameters.

## Coding variant associations

We also examined whether any of the lead variants were in strong LD ($r^2 > 0.8$) with coding variants ($p_{\text{discovery}} < 5 \times 10^{-8}$ and $p_{\text{replication}} < 0.05$).

We identified 10 common (MAF > 5%) loci harbouring coding variants or proxies (coding/splice region variants) in strong LD with lead variants (Supplementary Data 8). With the exception of the complex chr17:47 Mb (17q21.31, *MAPT*) locus, which contained coding/splice region proxies for multiple genes (*ARHGAP27, PLEKHM1, CRHR1, SPPL2C, MAPT, STH, KANSL1*), the other 9 loci contained coding variants affecting proxies for single genes. These included *ROR1* [rs7527017, Thr518Met], *THBS3* [rs35154152, Ser279Gly], *SLC6A20* [rs17279437, Thr199Met], *EPHA3* [rs35124509, Trp924Arg], *MSH3* [rs1650697, Ile79Val], *GNA12* [rs798488, start-lost], *PDGFRL* [rs2705051, splice region variant], *EML1* [rs34198557, Ala377Val] and *TSPAN10* [rs6420484, Tyr177Cys; rs1184909254/rs10536197, frameshift indel with stop codon gained]. Notably, the *SLC6A20* Thr199Met (rs17279437) variant, associated with widespread reductions in sulcal width (Supplementary Data 8), has previously been associated with reduced thickness of retinal components and with increased glycine and proline derivatives in CSF and urine, consistent with the role of *SLC6A20* as co-transporter regulating glycine and proline levels in the brain and kidneys (see Supplementary Information for details).

### Rare variant gene burden associations

We examined the impact of the burden of rare (MAF < 1%), loss of function protein-truncating variants (PTV) on sulcal measures across all 40,169 imaging participants. We found 50 PTV-burden sulcal measure associations at $p < 2.7 \times 10^{-6}$ (0.05/18,406 genes tested) but no significant associations after correcting for the number of sulcal measures ($p < 6.0 \times 10^{-9}$, 0.05/18,406/450 sulcal measures) (Supplementary Data 9). Of the 50 PTV-burden associations at $p < 2.7 \times 10^{-6}$, all but one were not observed at the same $p$-value threshold ($p < 2.7 \times 10^{-6}$) in the single variant primary associations in the same regions (+/−500 Kb). At the current sample size, the vast majority of genetic influences on sulcal measures came from low frequency (MAF > 1%) to common variants (MAF > 5%).

### Genetic and phenotypic correlations of brain folding

We investigated phenotypic and genetic correlation (GC) between measures from the right and left hemispheres as well as between different shape parameters of the sulcal measurements. We found high correlations between brain sulcal measurements across left and right sides, within and between the four shape parameters (Fig. 1c, Supplementary Fig. 6). In general, sulcal width measures showed the strongest correlations between left and right hemispheres compared to length, mean depth and surface area (Fig. 1d top). The high genetic correlation between hemispheres may explain the higher magnitudes of the association Z-scores of bilateral brain sulcal measures compared to hemisphere-specific analyses. We found average length, mean depth and surface area parameters to be positively correlated, with correlation between length and surface area being the strongest, and width to be negatively correlated with the other 3 shape parameters (Fig. 1d middle and bottom). Similar patterns of correlations between shape parameters were seen for left and right hemispheres as well as for both genetic and phenotypic correlations (Fig. 1d middle and bottom, Supplementary Fig. 6).

### Brain folding genes enriched for cortical expression and neurodevelopmental processes

To determine whether genes in the associated regions were enriched for expression in certain tissues, we performed enrichment analysis of annotated genes in significant loci ($p < 5 \times 10^{-8}$) for tissue gene expression in an independent dataset (Human Protein Atlas, Methods). We found significant enrichment of brain folding genes of approximately two-fold for expression in the cerebral cortex after multiple testing correction ($p = 7.3 \times 10^{-7}$). Using the most stringent replicated association threshold, $p < 2 \times 10^{-10}$, a similar two-fold enrichment for cerebral cortex expression remained significant ($p = 0.026$). The effect also remained significant with other sensitivity analysis thresholds (Fig. 2a), suggesting associated brain folding genes may have local effects. We also performed enrichment analysis for gene ontology (GO) processes and KEGG pathways. Notably, we found significant (FDR < 0.05) enrichment for various neurodevelopmental processes, including neurogenesis and a range of cellular GO biological processes including synapse development, neuronal path finding and axon guidance, among others (see Fig. 2b).

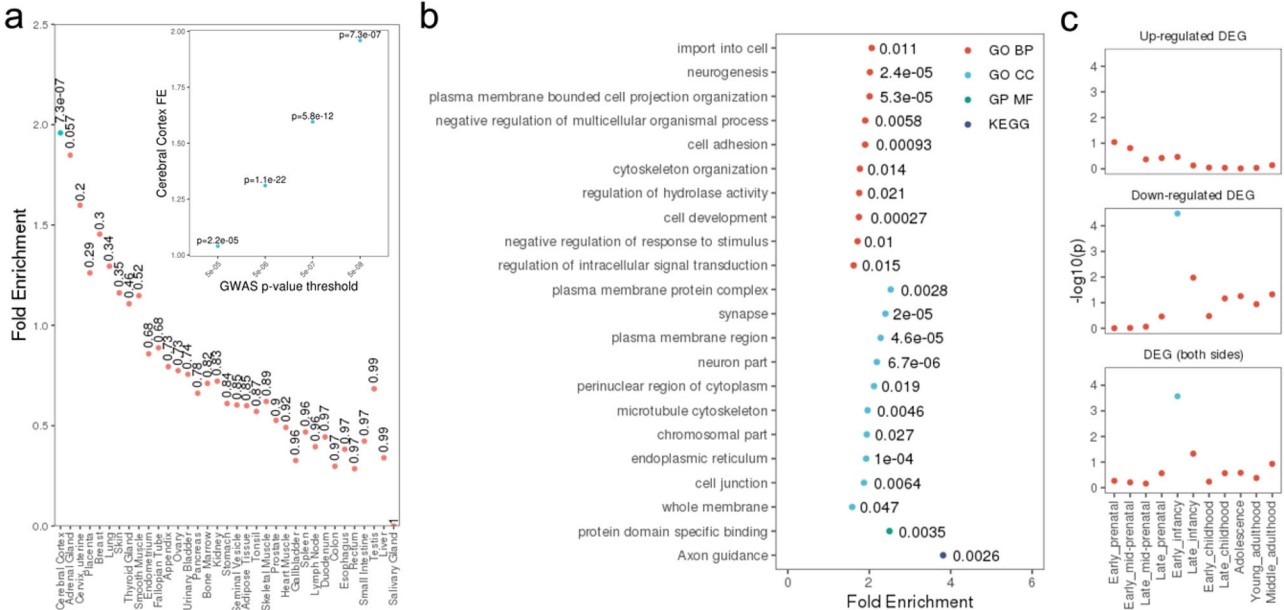

**Fig. 2 | Enrichment of genes in significant loci for tissue expression, pathways and brain developmental stages. a** Gene expression across various tissues (inset shows sensitivity analysis at other GWAS thresholds), blue points indicate multiple adjusted $p < 0.05$. Hypergeometric test used to derive $p$-values (uncorrected). **b** GO and KEGG pathways (FDR < 0.05). **c** Differentially expressed genes across brain development stages. Hypergeometric test used to derive $p$-values. Blue points indicate FDR < 0.05.

We also examined patterns of expression of the closest genes to each lead SNP using FUMA[25]. The extent and timing of gene expression across brain developmental stages was investigated using BrainSpan data[26]. This revealed a higher number of genes than expected (FDR < 0.05) were downregulated in early infancy compared to other developmental stages (Fig. 2c), driven by clusters of genes with higher expression levels during pre- compared to postnatal periods (Supplementary Fig. 7). These clusters include genes previously linked to neurodevelopment, with roles in the negative regulation of cell growth and proliferation, including *DAAM1*[27], *NT5C2*[28] and *NEO1*[29]. Using GTEx v8 data[30] significant downregulation of the closest gene set was seen across multiple adult tissues, including adult cortex (Supplementary Fig. 8a). Investigation of gene expression in embryonic cortical cells[31] revealed significant upregulation of expression in gestation week (GW) 10 stem cells (Supplementary Fig. 8b) and suggestive evidence of upregulation particularly for GW10 and GW16 microglia, although these were no longer significant following FDR correction.

## Colocalization with brain eQTLs to prioritise candidate genes

We performed colocalization analysis between brain cortical folding loci and the largest cortical expression quantitative trait locus (eQTL) summary dataset generated to date, Metabrain[32]. We found 27 of 119 loci to be colocalized for at least one sulcal measure, with one or more *cis* eQTLs in the cerebral cortex at a posterior probability (coloc PP4) > 0.7 and an additional 7 at a suggestive PP4 > 0.5 (Supplementary Data 10, Supplementary Fig. 9). A total of 53 unique cortical gene eQTLs colocalized (PP4 > 0.7) with at least one sulcal trait in the cortex. 15 of the 27 loci were colocalized with one unique eQTL in the cortex, 9 loci colocalized with 2 eQTLs, and the pleiotropic chr17:47 Mb *MAPT-KANSL1* locus colocalized with 14 different eQTL genes in a complex pattern (Supplementary Fig. 9, Supplementary Data 10). Across other brain-related tissues including the cerebellum, basal ganglia, hippocampus and spinal cord, we found a total of 25 loci in the cerebellum, 7 in the basal ganglia, 6 in the hippocampus and 3 in the spinal cord that colocalized (PP4 > 0.7) with at least one eQTL, with 9, 2 and 1 colocalized loci in the cerebellum, hippocampus and basal ganglia respectively, not found in cortex tissue.

## Multi-trait colocalization of cortex specific *KCNK2* eQTL and sulcal widths

The pleotropic chr1:215 Mb locus near *KCNK2* associated with multiple sulcal measures across the cortex in a largely symmetrical manner. The strongest lead variant ~40 Kb upstream of *KCNK2*, rs1452628:T, exhibited its strongest associations with reduced sulcal widths in superior brain regions (Fig. 3a, Supplementary Data 3 and 4). Notably, sulcal width associations at this locus showed evidence of co-localisation with cortex-specific *KCNK2* eQTLs from MetaBrain[32] (Supplementary Fig. 9, Supplementary Data 10), where rs1452628:T was associated with increased *KCNK2* expression in the cortex only (beta = 0.14, $p = 8.0 \times 10^{-7}$) (cf. cerebellum, hippocampus, basal ganglia and spinal cord, all $p > 0.1$, Fig. 3b, left). We formally tested whether regional sulcal width associations at this locus were driven by the same underlying variant influencing cortical *KCNK2* expression using the HyPrColoc multi-trait colocalization approach[24]. We found that all associations multi-colocalized to the same variant (posterior probability of colocalization = 0.74), with the candidate causal variant, rs1452628, explaining all of the posterior probability of colocalization (Fig. 3b right). We further assessed sensitivity to our choice of prior probability of colocalization. Joint colocalization across all or almost all of the traits remained, even after sequentially reducing the prior probability (Supplementary Information). These results suggest that a shared underlying variant drives all sulcal morphology associations and cortex-specific *KCNK2* expression at this locus.

## Genetic correlation between brain folding and neuropsychiatric phenotypes

Cross referencing with previous non-imaging trait and diseases in the GWAS Catalogue, we found that 56 of the 119 loci (at $p < 5 \times 10^{-8}$, 19 of 44 loci at $p < 2 \times 10^{-10}$) were associated with one or more diseases or intermediate phenotypes (Supplementary Data 11). We further investigated the genetic correlation (GC) of regional brain folding with 12 brain-related phenotypes, including neurological diseases, psychiatric illnesses and cognitive assessments (Methods, Supplementary Information). Using an empirical permutation threshold of $p < 0.0044$ to account for extensive correlations within brain folding phenotypes and neuro-related illnesses (Methods), we observed 158 significant GCs between regional brain folding measures and 10 distinct neuropsychiatric and cognitive phenotypes (Supplementary Data 12).

We examined GC between each of the four shape parameters, averaged across brain sulci, and neuropsychiatric phenotypes - observing at least two distinct clusters. Attention deficit hyperactive disorder (ADHD) and major depressive disorder (MDD) showed negative GCs with mean sulcal depth, length and surface area across multiple regions (Fig. 4a, b, Supplementary Data 12). Cognitive performance and Parkinson's disease (PD) showed positive GCs with sulcal length, surface area and depth (Fig. 4a, b, Supplementary Data 12). Focusing on specific brain sulci, we found the strongest GCs between PD and the length ($r_G = 0.40$, $p = 3.0 \times 10^{-3}$) and surface area ($r_G = 0.33$, $p = 6.0 \times 10^{-4}$) of the central sulcus (Supplementary Data 12), potentially indicating involvement of the sulcal folds adjoining the primary motor cortex in PD. We noted that sulcal width measures showed largely opposite directions of GC with neuropsychiatric phenotypes compared with sulcal depth, length or surface area (Fig. 4a), in keeping with their correlation structure (Fig. 1d).

## Interactive 3D visualisation of associations

Given the complexity and interdependencies of regional brain folding, visualising variant association results interactively in 3D may provide a more intuitive context to interpret association results, facilitating insights into genetic effects across multiple brain regions. We created an interactive resource (https://enigma-brain.org/sulci-browser) where users can query individual genetic variants and visualise their effects on sulcal width, depth, length and surface area across all regional brain folds interactively (Fig. 5).

For example: Visualising results for significant pleiotropic associations, such as chr12:106 Mb (*NUAK1*), chr16:87 Mb (near *C16orf95*) and chr6:126 Mb (containing *CENPW*), reveals how these loci affect multiple, widespread brain regions and shape parameters in distinct and complex ways (Fig. 5a–c). In contrast, visualising the chr15:40 Mb (15q14) locus associations, mostly tagged by rs4924345, reveals how these effects are more localised (Fig. 5d). More specifically, we observed strong *positive* effects of the minor rs4924345:C allele on bilateral central sulcus mean depth (beta$_{dis}$ = 0.29, $p_{dis} = 3.1 \times 10^{-79}$) and surface area (beta$_{dis}$ = 0.15, $p_{dis} = 6.0 \times 10^{-25}$) but *negative* effects bilaterally on neighbouring superior postcentral intraparietal superior sulcus mean depth (beta$_{dis}$ = −0.14, $p_{dis} = 1.0 \times 10^{-18}$) and surface area (beta$_{dis}$ = −0.11, $p_{dis} = 6.5 \times 10^{-13}$); retro central transverse ramus of the lateral fissure mean depth (beta$_{dis}$ = −0.16, $p_{dis} = 7.9 \times 10^{-21}$) and surface area (beta$_{dis}$ = −0.15, $p_{dis} = 2.8 \times 10^{-18}$); inferior precentral sulcus mean depth (beta$_{dis}$ = −0.14, $p_{dis} = 5.6 \times 10^{-16}$), surface area (beta$_{dis}$ = −0.16, $p_{dis} = 8.0 \times 10^{-19}$) and length (beta$_{dis}$ = −0.11, $p_{dis} = 1.3 \times 10^{-9}$).

This sulcal visualisation tool can provide renderings based on effect sizes, Z-scores or *p*-values, as well as an option to download query results.

## Discussion

Cortical morphogenesis is an orchestrated, multifaceted process that shows striking consistency across individuals[33]. The formation of the brain's characteristic convex folds (gyri) and valleys (sulci) is regulated

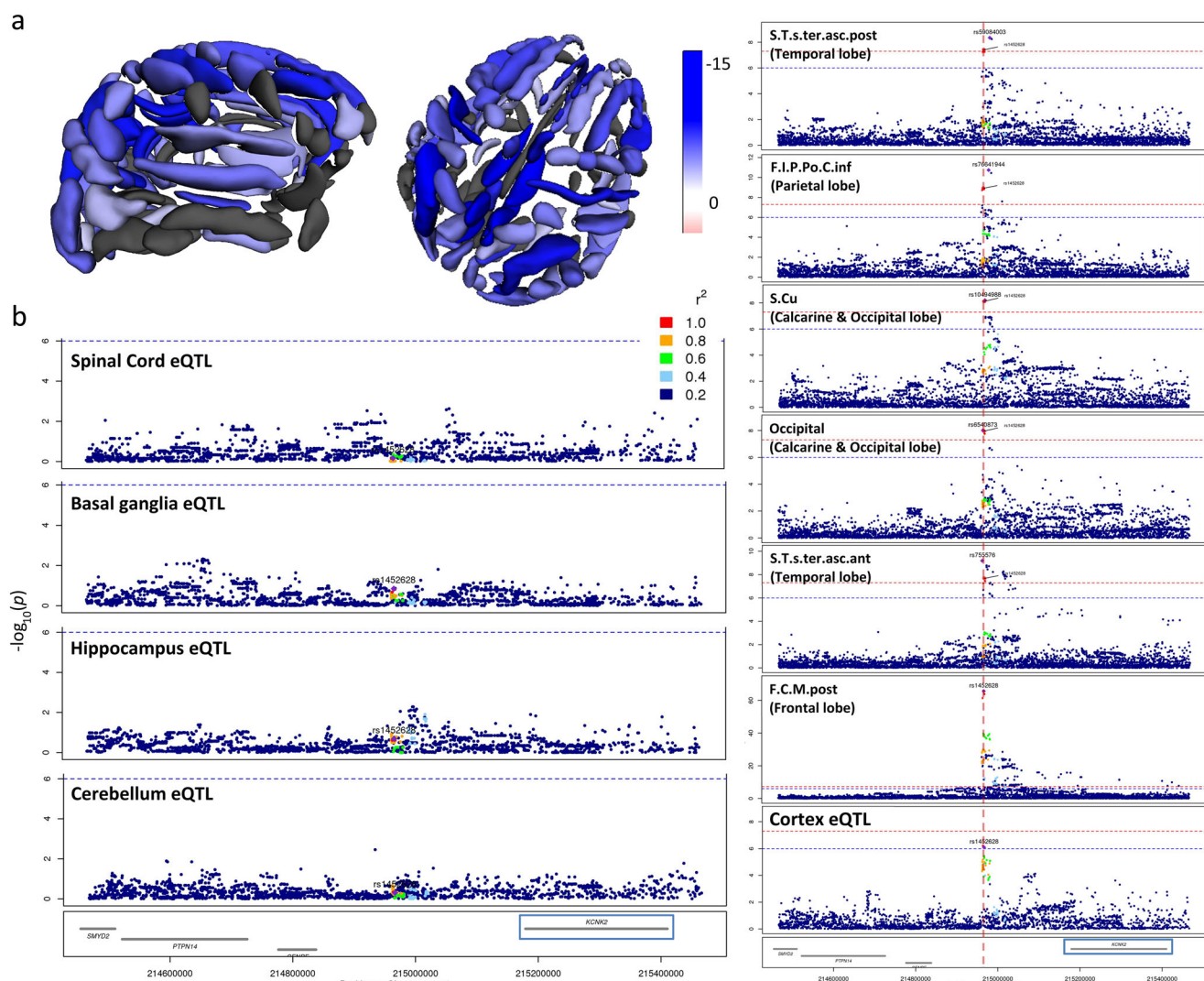

**Fig. 3 | *KCNK2* locus associations. a** Association of the lead rs1452628:T variant with reduced sulcal widths across the brain. (Grey colours indicate associations with $p_{rep} > 0.05$). **b** Left: regional association plot of MetaBrain *KCNK2* eQTLs for spinal cord, basal ganglia, hippocampus and cerebellum. Right: regional association plots and colocalization of cortex *KCNK2* eQTL and different lead variants in the *KCNK2* locus. A subset of associations shown for each different lead variant shown due to space constraints. *P* derived from regression-based tests.

by a complex interplay of cellular, biomechanical and genetic influences[9], but our understanding of its genetic underpinnings has been limited[34,35]. Recent investigations have revealed insights into the genetic architecture of sulcal depth[15] but genetic factors influencing other important and heritable features, such as sulcal width, depth and surface area, have not been examined at comparable scale[16,17]. Measures of sulcal morphology can serve as sensitive markers of aging[21,22,36] and intelligence[37], and effects on these measures are partially independent of those on cortical thickness or surface area[38]. Thus, studying genetic influences on brain sulci may complement investigations of more traditional structural measures, offering greater understanding of the mechanisms guiding variation in human brain organisation and downstream associations with human health and disease[17].

Here, combining densely-imputed genetic variants with whole-exome sequencing, we performed the most comprehensive genetic mapping of regional cortical sulcal morphometry to date, identifying 119 unique genetic loci influencing human sulcal depth, width, length and surface area. We discovered over 60 novel loci not previously implicated in any brain imaging-related association studies. The number of genetic associations observed across different sulcal parameters was approximately in accordance with their heritability[17]. We

observed stronger genetic correlations than phenotypic correlations between left and right sides, suggesting that environmental and non-genetic factors may play a role in structural and functional lateralisation. Notably, regional sulcal width measures clustered in a manner that reflected broad brain topology, potentially underlining strong prenatal influences on sulcal development[39–42], and the relatively higher heritability of sulcal width versus sulcal depth, length, or surface area[17]. We note that the vast majority of observed associations were driven by low frequency (MAF > 1%) to common variants (MAF > 5%) well-covered by genotyping with imputation. Therefore, at the current sample sizes, single rare (MAF < 1%) variants and aggregated burden of rare PTVs provide limited additional benefits to imputation for detection of genetic influences on brain sulcal morphology. Potential rare variant associations with large effects could still be detected at increased sample sizes.

We demonstrated the highly polygenic genetic architecture of brain folding, which has both local and widespread effects within the brain. When visualised in 3D, local effects are apparent, that are likely to be missed in globally aggregated brain measurement studies. We also implicated specific candidate genes in several cases through coding variants in LD. We added exonic resolution through WES, as

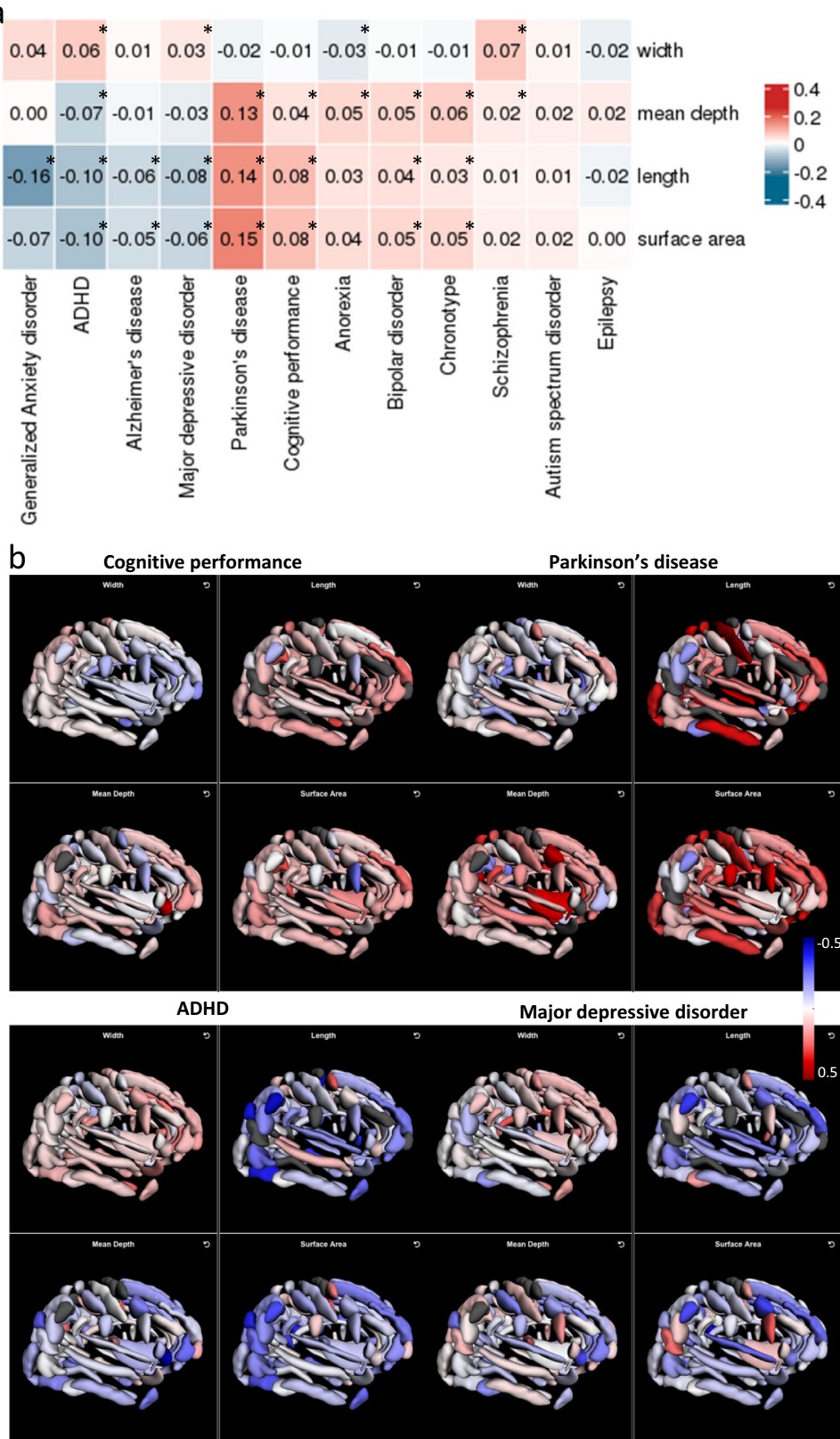

**Fig. 4 | Genetic correlations with neuropsychiatric conditions. a** Genetic correlations between shape parameters and neuropsychiatric conditions. Genetic correlations (GCs) were averaged across all brain regions for each sulcal parameter separately – the mean GCs are displayed in each entry. * indicate *p* < 0.001 (adjusted for number of tests) derived from two-sided *t*-tests. **b** Examples of genetic correlations across brain sulcal folds with cognitive performance, Parkinson's disease, attention deficit hyperactive disorder (ADHD) and major depressive disorder.

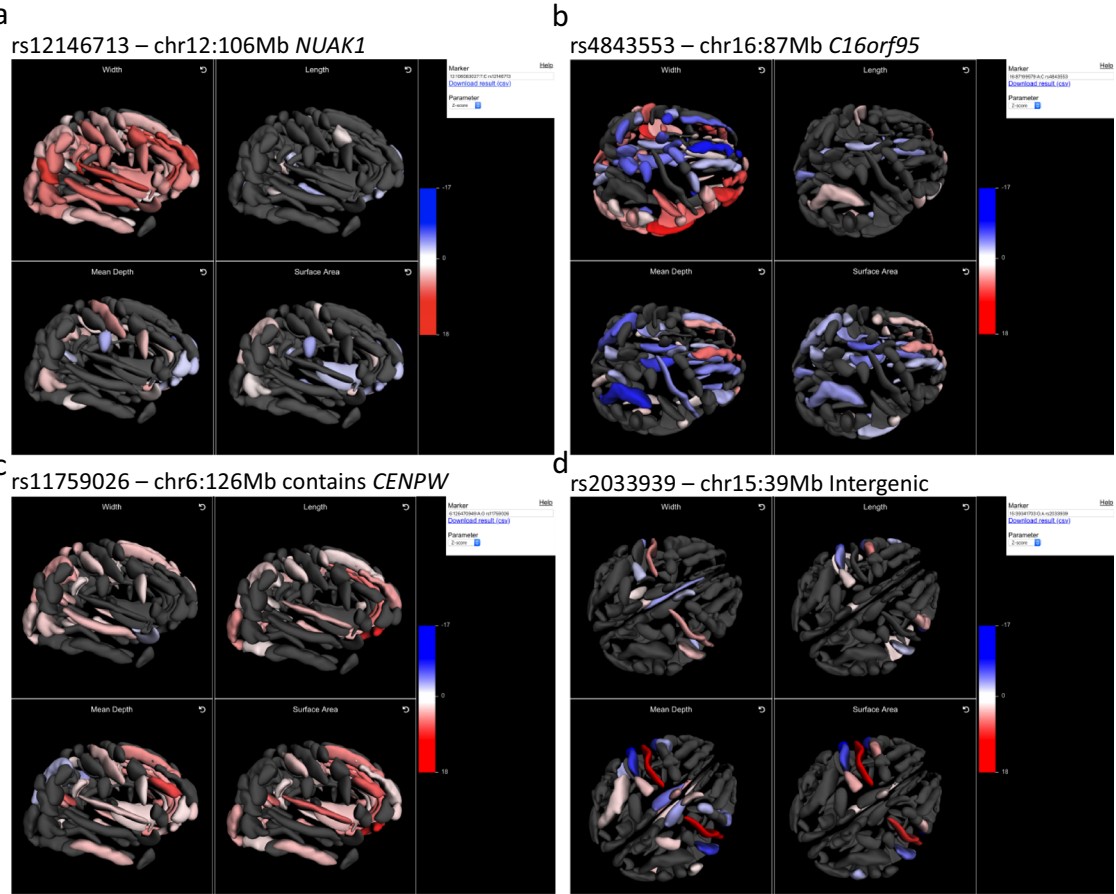

**Fig. 5 | Three-dimensional visualisation of brain sulcal associations (Z-scores) for four exemplar pleiotropic loci.** An interactive online tool, available at https://enigma-brain.org/sulci-browser, was used to visualise genetic influences on sulcal morphology in three dimensions, revealing (**a**) widespread, positive associations between rs12146713 (*NUAK1*) and sulcal width measures, (**b**) widespread positive and negative associations between rs4843553 (near *C16orf95*) and sulcal width, depth and surface area measures, (**c**) widespread positive associations between rs11759026, containing *CENPW*, and all four sulcal measures, and (**d**) bilateral associations between an intergenic variant, rs2033939, and length, width, depth and surface area of the central sulci, precentral sulci, and posterior lateral fissures.

well as through colocalization with brain eQTLs using a large-scale brain-specific dataset for better power and specificity[32]. We observed pleiotropic associations at genetic loci consistently implicated in prior genetic studies of neuroimaging phenotypes, such as the *MAPT-KANSL1* locus[43,44], while resolving other associations to specific brain regions and sulcal folding parameters, such as the *KCNK2* locus and sulcal width. We also show that the majority of the associations at pleiotropic loci co-localise to one or two shared signals, suggesting largely homogeneous effects at these loci. The *MAPT-KANSL* has an extensive heterogeneous pleiotropy, in keeping with the complexity at this locus[45].

Our results provide evidence of enrichment of associated genes for expression in the cerebral cortex, strongly implicating genes involved in neurodevelopment. We found enrichment for differential gene expression occurring in early brain development, indicating that genetic effects on cortical gyrification occur most prominently during early life, likely via modulation of neurodevelopmental pathways. Inherited functional impairments of these genes and their associated pathways may increase the risk for neurodevelopmental disorders. For example, homozygous and compound heterozygous mutations at *EML1* - a gene associated with right insula surface area - cause band heterotopia, a neuronal migration disorder characterised by intellectual disability and epilepsy[46]. Similarly, heterozygous deletion of *ZIC1* and *ZIC4* is associated with Dandy-Walker malformation, a congenital cerebellar malformation[47], whereas contiguous deletions at the 16q24.3 locus encompassing *CENPW* cause

microcephaly, distichiasis, vesico-ureteral and intellectual impairment[48]. Additionally, genetic variants at *NUAK1* - a pleiotropic locus associated with frontal, temporal and precentral sulcal widths - have shown links to autism spectrum disorder[49,50], ADHD[51] and cognitive impairment[52].

Globally, genetic variants influencing cortical gyrification showed robust, widespread correlation with variants influencing cognitive performance, schizophrenia, ADHD and depression, suggesting a shared molecular system potentially underpinning neurodevelopmental and neuropsychiatric disorders[53,54]. The strongest correlations were observed for localised sulcal measures, such as those between anterior inferior temporal sulcus length and ADHD, warranting further investigations in independent datasets. We also observed significant positive genetic correlations between Parkinson's disease (PD) and various sulcal traits – most notably, with bilateral calcarine and occipital lobe surface area and length – contrasting with significant negative correlations between Alzheimer's disease and the same sulcal measures. These findings expand upon prior reports of positive genetic correlations between PD and cortical surface area[55]. Our results may indicate divergent genetic contributions to cortical thickness and cortical surface area in PD, whereby increased cortical surface area reflects genetic influences on neural progenitor differentiation, defining the number of neocortical columns, during embryogenesis, whereas reduced cortical thickness reflects events later in development, influencing the number of synapses and neurons per neocortical column[56,57].

Through multi-trait colocalization, we identified a shared underlying genetic driver of increased cortical *KCNK2* expression and pleiotropic effects on reduced sulcal widths. KCNK2, also known as TREK-1, is a two-pore domain potassium channel highly expressed in the central nervous system and modulated by both chemical and physical stimuli[58,59]. KCNK2 regulates immune-cell trafficking into the CNS[58] and genetic ablation of *Kcnk2* is associated with neuroinflammation, blood-brain barrier impairment[60] and increased sensitivity to ischaemia and epilepsy in mice[61]. In addition to brain volume, the *KCNK2* locus was previously implicated in sulcal opening[16] and the same lead variant, rs1452628:T, was associated with difference between predicted brain age and chronological age[62]. Our findings re-emphasize the role of *KCNK2* in cerebral cortex development, alongside similarly pleiotropic and widely-investigated therapeutic targets such as *NUAK1*[63] and *MAPT*[64]. Further investigation of the links between these proteins and disease processes downstream of cortical gyrification may support therapeutic development.

One notable limitation of the present study is that genetic associations were identified in a population of mostly British individuals. Additionally, dividing UK Biobank participants into discovery and replication cohorts prioritised robustness of genetic associations, but reduced power to detect rare and low frequency variant associations. Larger sample sizes will increase power and refine the estimates reported here. Our method to ascertain brain folding phenotypes is applicable across different MRI scanning protocols, which vary across sites[17]. This should facilitate large-scale, cross-biobank studies of cortical folding and minimise site- and cohort-specific effects. Notably, our study shows partial overlap with a recent genetic investigation of sulcal depth in the UK Biobank[15]. Although the cohorts under investigation are comparable, our study focuses on four sulcal descriptors instead of one, integrates a greater number of rare variants via WES, incorporates finer-grained brain expression datasets with downstream colocalization, and provides a more regionalised genetic investigation, based on an established probabilistic atlas of brain sulci[65]. Finally, we note that while sulcal characteristics form distinct phenotypic clusters compared with pre-extracted measures of cortical thickness and surface area, all novel associations reported in this study should not be interpreted as unique genetic influences on sulcal morphology, given that larger-scale investigations may reveal overlapping associations with independent brain imaging measures.

To aid interpretation and increase the utility of our results to the wider scientific community, we created an interactive 3D visualisation of our associations, where users can query specific variant associations across the entire brain and the shape parameters simultaneously. We highlighted various cases where complex and pleiotropic associations differ in brain region and shape parameter distributions, which become more apparent when represented visually in three dimensions. However, care is needed to avoid over-interpreting weaker and more heterogeneous associations.

In conclusion, we provide the most comprehensive genetic atlas of regional brain folding to date, identifying novel associations and insights into processes that drive the genetic effects, as well as providing a resource for the wider community for further elucidation of specific findings.

## Methods

### Samples and participants

UK Biobank (UKB) is a UK population study of ~500,000 participants aged 40–69 years at recruitment[66]. Participant data include genomic, imaging data, electronic health record linkage, biomarkers, physical and anthropometric measurements. Further details are available at https://biobank.ndph.ox.ac.uk/showcase/. Informed consent was obtained from all participants. Analyses in this study were conducted under UK Biobank Approved Project numbers 26041 and 11559.

### Brain folding imaging phenotypes

The UK Biobank began collecting brain MRI scans in 2014 with the goal of scanning 100,000 individuals. The protocol includes isotropic 3D T1-weighted (T1w) MP-RAGE images (voxel size 1 mm$^3$; field-of-view: $208 \times 256 \times 256$) that have undergone bias-field correction in the scanner. Full acquisition details can be found in[67]. T1w images were processed using FreeSurfer (v7.1.1) and quality controlled using protocols developed by the Enhancing Neuro Imaging Genetics for Meta-Analysis (ENIGMA) consortium (http://enigma.ini.usc.edu/). BrainVISA (http://brainvisa.info) was implemented for sulcal classification and labelling[65,68]. Prior applications of BrainVISA to study human brain development and disease are summarised in Supplementary Information. Morphologist 2015, an image-processing pipeline included in BrainVISA, was used to measure sulcal shape descriptors. To improve sulcal extraction and build on current protocols used to analyse thousands of brain scans, quality controlled FreeSurfer outputs (*orig.mgz, ribbon.mgz and talairach.auto*) were directly imported into the pipeline to avoid re-computing intensities inhomogeneities correction and grey/white matter classification. Sulci were then automatically labelled according to a predefined anatomical nomenclature[68,69]. This protocol is part of the ENIGMA-SULCI working group; a Docker and a Singularity container have been created to facilitate the processing on computational clusters (https://hub.docker.com/repository/docker/fpizzaga/sulci). We retained length, width, depth, and surface area for all 121 sulcal measurements derived from this protocol for a total of 484 phenotypes (https://surfer.nmr.mgh.harvard.edu/)[68,69].

Phenotypes with missingness >75% were excluded from subsequent analysis, leaving 450 measurements (224 left and 225 right hemisphere measures) for analysis. Missingness occurs mainly with smaller sulci that are not identified in some individual MRIs. Prior to analysis, all imaging phenotypes were inverse-rank normalised to approximate a standard normal distribution and minimise effects of outliers.

Non-linear dimension reduction method, t-distributed stochastic neighbour embedding (t-SNE) was used to maps the high-dimensional brain imaging phenotypes to two dimensions whilst preserving local structure, such that close neighbours remain close and distant points remain distant[23]. This approach has been widely used in high-dimensional data (such as transcriptomics), compared to principal component analysis, which is a linear dimension reduction method that aims to preserve global (*cf* local) structure[70]. T-SNE often performs better on high-dimensional data in revealing local structures[23,70]. We performed t-SNE on inverse-rank normalised imaging phenotypes (after imputation for missing data using imputePCA function from the missMDA R package[71]) for the combined sulcal measures and existing UKB structural brain imaging phenotypes (T1 structural brain MRI, T2-weighted brain MRI, diffusion brain MRI, $n = 2125$, Supplementary Data 2), all sulcal measures, as well as separately within each sulcal parameter.

### Discovery and replication cohorts

We partitioned UKB samples with MRI measurements into discovery and replication approximately in 2:1 split. The discovery cohort were comprised of MRI measures in individuals of European ancestry from Newcastle, Cheadle and Reading imaging centres, whilst the replication cohort composed of the remaining (non-European) individuals from the aforementioned three centres, and mostly all individuals from the Bristol imaging centre. Subsequent analyses were performed treating the discovery and replication cohorts as completely separate to minimise data contamination and biases.

### Genetic data processing

**UKB genetic QC**. UKB genotyping and imputation (and QC) were performed as described previously[66]. WES data for UKB participants

were generated at the Regeneron Genetics Center (RGC) as part of a collaboration between AbbVie, Alnylam Pharmaceuticals, AstraZeneca, Biogen, Bristol-Myers Squibb, Pfizer, Regeneron and Takeda with the UK Biobank[72]. WES data were processed using the RGC SBP pipeline as described in[73,74]. RGC generated a QC-passing "Goldilocks" set of genetic variants from a total of 454,803 sequenced UK Biobank participants for analysis. Additional QC were performed prior to association analyses as detailed below.

**Additional QC and variant processing.** In addition to checking for sex mismatch, sex chromosome aneuploidy, and heterozygosity checks, imputed genetic variants were filtered for $INFO > 0.8$, $MAF > 0.01$ (rarer variants around coding regions would be better captured by WES) globally across UKB and chromosome positions were lifted to hg38 build. WES variants were filtered for $MAC > 10$ within the UKB subset with MRI measurements. Imputed and WES variants were combined by chromosome position (hg38) and alleles and in the case of overlaps, the WES variant was retained (as WES generally have higher quality calls compared to imputation). Variant annotation was performed using VEP[75] with Ensembl canonical transcripts used where possible.

**Genetic association analyses.** GWAS were performed using REGENIE v2.0.1 via a two-step procedure to account for population structure detailed in[76]. In brief, the first step fits a whole genome regression model for individual trait predictions based on genetic data using the leave one chromosome out (LOCO) scheme. We used a set of high-quality genotyped variants: minor allele frequency $(MAF) > 1\%$, minor allele count $(MAC) > 100$, genotyping rate $>99\%$, Hardy-Weinberg equilibrium (HWE) test $p > 10^{-15}$, $<10\%$ missingness and linkage-disequilibrium (LD) pruning (1000 variant windows, 100 sliding windows and $r^2 < 0.8$). The LOCO phenotypic predictions were used as offsets in step 2 which performs variant association analyses using standard linear regression. We limited analyses to variants with $MAC > 50$ to minimise spurious associations. The association models in both steps also included the following covariates: age, $age^2$, sex, age*sex, $age^2$*sex, imaging centre, intracranial volume, first 10 genetic principal components (PCs) derived from the high-quality genotyped variants (described above) and additionally first 20 PCs derived from high-quality rare WES variants $(MAF < 1\%, MAC > 5$, genotyping rate $>99\%$, HWE test $p > 10^{-15}$, $<10\%$ missingness) as additional control for fine-scale population structure.

**Definition and refinement of significant loci.** To define significance, we used multiple testing corrected threshold of $p < 2 \times 10^{-10}$ ($5 \times 10^{-8}$/ 273 approximate number of independent trait). We used phenotypic PCs accounting for 90% of phenotype variance to estimate the approximate number of independent traits to account for correlations between regions, side and parameters. Additionally, we also require at least nominal significance $(p < 0.05)$ with concordant directions in the replication cohort which should limit false positives even at $p < 5 \times 10^{-8}$. For reporting, we also included the standard genome-wide significant loci $(p < 5 \times 10^{-8})$ that replicated at $p < 0.05$ in the replication cohort.

We defined independent trait associations through clumping ±500 Kb around the lead variants using PLINK[77], excluding the HLA region (chr6:25.5-34.0 Mb) which is treated as one locus due to complex and extensive LD patterns. As overlapping genetic regions may be associated with multiple correlated measurements and to avoid over-reporting genetic loci, we merged overlapping independent genetic regions (±500 Kb) across traits and collapsed them into one locus.

**Rare variant burden analyses.** We investigated the impact of rare loss of function variant burdens on sulcal measures across the entire 40,169 imaging cohort. Protein-truncating variants (PTVs) were defined using VEP v96 and LOFTEE[78]. LOFTEE applies a range of filters on stop-gained, splice-site disrupting and frameshift variants to exclude putative PTVs due to variant annotation and sequencing mapping errors that are unlikely to substantially disrupt gene function. We extracted variants predicted as PTVs with 'high confidence' by LOFTEE for 18,406 genes. Burden testing was performed with variant MAF cut-offs of 1%, 0.1%, 0.01% and singleton, in REGENIE as part of the step 2 procedure with the same covariates as the single variant analyses.

**Cross reference with known genetic associations.** We cross-referenced the lead variants and their proxies (LD proxy $r^2 > 0.8$, $+/-500$ Kb around the lead variant, with HLA region treated as one region) for significant associations $(p < 5 \times 10^{-8})$ in GWAS Catalogue[6]. Brain imaging studies were separated from other intermediate and disease phenotypes as defined by the list of brain imaging studies in Supplementary Information.

**Expression enrichment.** We examined whether genes within associated loci are enriched for expression the various brain tissues. Enrichment analysis was performed using the TissueEnrich R package[79] using the annotated genes (available canonical genes mapped in VEP) for all genome-wide significant variants $(p < 5 \times 10^{-8}$, additional sensitivity analysis thresholds of $p < 5 \times 10^{-7}$, $5 \times 10^{-6}$, $5 \times 10^{-5}$ were used for cortex) and a background of annotated genes for all variants analysed. Specifically, we used the RNA dataset from Human Protein Atlas using all genes that are found to be expressed within each tissue.

**GO and KEGG process enrichment.** Using the same significant annotated genes and backgrounds as for the expression enrichment analyses, we performed enrichment testing for GO and KEGG pathways using the WEB-based GEne SeT AnaLysis Toolkit (WebGestalt)[80] (http://www.webgestalt.org/). We used the over-representation analysis method, analysing GO Biological Process, GO Cellular Component, GO Molecular Function and KEGG, with Benjamini-Hochberg FDR threshold of 0.05 for significance. We used the default parameters of minimum of 5 and maximum 2000 genes per category. Related process and pathway entries were grouped through the inbuilt weighted set cover redundancy reduction approach.

**FUMA analyses of expression timing.** The closest genes to each lead SNP were annotated using MAGMA[81], and this set of closest genes used in all FUMA analyses. Gene expression timing and tissue enrichment was assessed for the set of genes closest to each lead SNP using the GENE2FUNC function, based on averaged log2 transformed expression levels compared across each label (i.e. Brainspan brain age for the expression enrichment analysis, and GTEx tissue types for the tissue specificity analysis). Gene sets are defined as differentially expressed when the Bonferroni corrected $p \leq 0.05$ and the absolute log fold change $\geq 0.58$ between a specific label (brain age or tissue type) compared to others[25]. All other annotated genes/transcripts in each dataset were included as background genes for comparison in hypergeometric tests of the 'closest gene' set. Significantly enriched gene sets had FDR corrected $p \leq 0.05$. All other annotated genes/transcripts in the BrainSpan data were included as background genes for comparison in hypergeometric tests of gene sets. Significantly enriched gene sets had FDR corrected $p < 0.05$.

Cell-type specificity analyses were conducted using human embryonic prefrontal cortex single-cell RNA expression data (normalised as the number of specific transcript reads per million transcripts per cell) generated by Zhong et al. (2018)[31], to investigate expression of the closest gene set across the foetal neurodevelopmental period. As above significant enrichment of a gene set for a specific label (cell type per age) is indicated by an FDR-corrected $p \leq 0.05$.

**Genetic correlation analysis.** We performed genetic correlation analysis between brain folding phenotypes (including hemispheres and shape parameters), and 12 neuropsychiatric conditions with readily available summary data using LD score regression (LDSC v1.0.1)[82]. We also performed SNP-based heritability estimation using LDSC. Genetic variants were filtered and processed using the "munge_sumstats.py" in LDSC and we used LD scores recommended by the software authors[82].

To account for multiple testing of extensive related and correlated phenotypes, we permuted each neuropsychiatric condition Z-score 100 times (limited by computational cost) and tested each permuted neuropsychiatric condition with each brain folding phenotype to generate an empirical multiple testing threshold of $p = 0.0044$ (approximately adjusted $p < 0.01$ from 100 permutations).

**Colocalization analyses.** We performed colocalization analyses[83] between brain eQTLs from MetaBrain and brain folding loci using the coloc R package. We used the default priors ($p1 = 10^{-4}$, $p2 = 10^{-4}$, $p12 = 10^{-5}$) with regions defined as +/−500 Kb around the lead variant. Evidence for colocalization was assessed using the posterior probability (PP) for hypothesis 4 (PP4; an association for both traits driven by the same causal variant). PP4 > 0.5 were deemed likely to colocalize as it guaranteed that hypothesis 4 was computed to have the highest posterior probability, and PP4 > 0.7 were deemed highly likely to colocalize.

To assess whether all traits jointly colocalize at the *KCNK2* locus (with brain eQTLs) and loci associated with multiple sulcal measures we used the multi-trait colocalization software HyPrColoc[24], using the recommended default settings and priors (HyPrColoc's default prior parameters $p = 10^{-4}$ and $p_c = 2 \times 10^{-2}$ are equivalent to setting $p1 = 10^{-4}$, $p2 = 10^{-4}$, $p12 = 2 \times 10^{-6}$ in coloc, hence the default prior probability of colocalization p12 is slightly more conservative than in coloc). HyPrColoc computes evidence supporting one or more clusters of traits colocalizing at a single variant in the region, concluding that a cluster of traits colocalize if the posterior probability of colocalization (PPC) is above a user defined threshold (PPC > 0.5 by default, which is equivalent to setting the algorithms' regional, $P_R$, and alignment, $P_A$, thresholds to 0.7 respectively). We also performed additional sensitivity analysis across different parameter specifications (Supplementary Information). Heatmaps produced using the ComplexHeatmap R package.

### Reporting summary

Further information on research design is available in the Nature Research Reporting Summary linked to this article.

### Data availability

The online browser for visualisation of results and links to summary association data and is available at https://enigma-brain.org/sulci-browser. Other datasets used in this study include Human Protein Atlas (https://www.proteinatlas.org/). BrainSpan (https://www.brainspan.org/). MetaBrain (https://www.metabrain.nl/) and GTEx (https://gtexportal.org/home/).

### Code availability

Codes used are part of standard software and tools. Additional details available in Methods.

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

## Acknowledgements

We thank all the participants, contributors and researchers of UK Biobank for making data available for this study. We thank the UK Biobank Exome Sequencing Consortium (AbbVie, Alnylam Pharmaceuticals, AstraZeneca, Biogen, Bristol-Myers Squibb, Pfizer, Regeneron and Takeda) for generation of the whole-exome sequencing data and Regeneron Genetics Centre for initial quality control of the exome sequencing data. We thank Peter Kochunov and his team for hosting the interactive browser, with support from NIH instrumentation grant S10OD023696. N.J. is supported by NIH grant R01AG059874. S.E.M. and J.N.P. are supported in part by NHMRC grants APP1172917 and APP1158127.

## Author contributions

S.J.L. and F.P. contributed equally to this work. Study conceptualisation and design: C.D.W., P.M.T., N.J., B.B.S.; methodology: B.B.S., S.J.L., P.M.T., N.J., C.D.W.; sulcal imaging processing: F.P., A.Z., D.D., T.I., I.B.G., N.J.; phenotype harmonisation: M.J., D.G.M., S.S.C., Biogen Biobank Team; analysis: B.B.S., S.J.L., J.N.P., S.E.M., C.N.F.; interactive browser: N.S., F.P.; writing: B.B.S., C.D.W., P.M.T., N.J., H.R.; all authors critically reviewed the manuscript.

## Competing interests

The authors declare the following competing interests: B.B.S., S.J.L., Biogen Biobank Team, M.J., D.G.M., H.R., C.D.W. are employees of Biogen. P.M.T and N.J received grant support from Biogen for this work. The remaining authors declare no competing interests.

## Additional information

## Biogen Biobank Team

**Benjamin Sun[1], Ellen Tsai[1], Paola Bronson[1], David Sexton[1], Sally John[1], Eric Marshall[1], Mehool Patel[1], Saranya Duraisamy[1], Timothy Swan[1], Dennis Baird[1], Chia-Yen Chen[1], Susan Eaton[1], Jake Gagnon[1], Feng Gao[1], Cynthia Gubbels[1], Yunfeng Huang[1], Varant Kupelian[1], Kejie Li[1], Dawei Liu[1], Stephanie Loomis[1], Helen McLaughlin[1], Adele Mitchell[1], Stephanie J. Loomis[1,10], Christopher D. Whelan ⓘ[1,11] ✉ & Heiko Runz ⓘ[1]**

