## [Peer Review File · Nature Communications]

Genetic map of regional sulcal morphology in the human brain from UK biobank dataEditorial Note: This manuscript has been previously reviewed at another journal that is not operating a transparent peer review scheme. This document only contains reviewer comments and rebuttal letters for versions considered at *Nature Communications*.

REVIEWER COMMENTS

Reviewer #4 (Remarks to the Author):

The work entitled 'Genetic map of regional sulcal morphology in the human brain' describes the genetic influences on sulcal morphology by using both genome-wide genotyping and whole exome sequencing data on a large UK biobank cohort of more than 40k participants. I commend authors for the solid analytic pipeline designed for the identification of genome wide significant loci (with discovery – replication cohorts), as well as for the permutation procedures to ensure low rate of false positive findings. The selected p-value threshold of 2×10^{-10} accounts for the multiple phenotypes tested and adds credibility to the results. Results are noteworthy, many tools have been used and methodology is solid.

The greatest novelty of the work relies on the discovery and functional characterization of new genetic loci involved in sulcal morphology, which has not been as extensively described as other brain morphology measures such as volume or cortical thinning. Functional and gene expression enrichment analysis across brain tissues, genetic/phenotypic correlations and co-localization analyses give strength and enrich the results. Overall, I think this is potentially a very valuable paper. However, I find the paper sometimes difficult to follow. I have some important concerns related with the genetic characterization of the GWAS results that need to be addressed and some minor comments:

Major concerns

- I think the weakest point of the paper is the rare variant analysis. What is the advantage provided by using the variants from WES in the association and replication analyses? If there is no clear benefit, I think it would be useful for future studies to state that genotype data can be sufficient to describe the biological underpinnings of similar or related to sulcal phenotypes.
- I think the study lacks a clear rare variants analysis, and I am not very convinced of what is the main contribution of the “coding variant association” section. While using a replication cohort provides a great robustness for the found results (as also mentioned in discussion), it generates a limitation for the detection of rare and private mutations. In the line of the above commented, burden tests for ultra-rare mutations collapsed on specific genes even using the whole cohort could explain some of the gene-based associations with sulcal phenotypes at the rare variation level. Since many neuropsychiatric phenotypes have dense differences between contributions from rare and common genetic variants, it will be interesting to see whether genetic influences on sulcal morphology also differ between rare and common variation. Also, results from rare predisposing variant accumulation tests could illustrate some of the related previous findings cited in the discussion paragraph (line 354 - 363).
- When interpreting gene enrichment across neurodevelopmental stages from BrainSpan, only genes downregulated during early infancy are overrepresented. However, instead of focusing on this finding and suggesting reliable explanations, authors focused on some of the genes putatively downregulated during early prenatal stages: DAAM1, NT5C2 and NEO1. These genes, however, seem to reach these highest expression levels at early prenatal stages (color at extended data figure 7 towards brown-red). Also, other genes from the significant list display the opposite pattern in comparison with that observed for the mentioned genes, and this is not described along the manuscript. Therefore, the sentence “These results suggest that genetic effects on regional brain folding are in part driven via regulation of neuronal development during early brain development” is not well demonstrated. This part is confusing and must be revised.

- The gene expression enrichment analyses could even be more refined including some cell-type specific gene-sets, with many recent databases having great information at this respect (see for instance doi.org/10.1093/nar/gkaa76). In the same line, FUMA can also assess for expression enrichment across brain tissues but these results are not reported. Have these results been checked? Are these results similar to those obtained with TissueEnrich R package?

- The authors focused on KCNK2. Is the multi co-localization described at this locus the norm or a specific case of this phenomenon? Why locus 12q23.3 harboring NUA1 gene is not tested when this gene is also highlighted at the results section? In line 144: "Notably, the chr1:215Mb locus (near KCNK2) and chr12:106Mb locus (12q23.3, NUA1) were associated with 23 and 22 width measures respectively across multiple brain regions"

- I miss more discussion related to the genetic correlations between brain sulcal morphology parameters and neuropsychiatric conditions. Although some interesting correlations arise (Parkinson disease or ADHD) and differential patterns across two observable clusters in relation to sulcal width are described, less is said about some intriguing differences. For instance, Schizophrenia and bipolar disorder, with one of the highest SNP-based co-heritability estimates, show opposite correlation with sulcal width. Any explanation for this? Is this a notable difference when comparing with correlation observed with other brain morphology traits? Other notable differences can be seen at Figure 4 between Alzheimer and Parkinson. Also, beyond differential patterns observed, these genetic correlation results could be useful to discuss with neuropsychiatric traits seem to be more impacted by variability at sulcal morphology.

Minor comments

- From the beginning of the results section, I understand that the 450 independent sulcal phenotypes selected for GWAS come from the dimensionality reduction with tSNE. The sentence at methods line 457 could be also included in results section to help follow the procedure.

- Figure 1C captures a lot of valuable information but it is very difficult to appreciate in full detail. Since this figure represents one of the main findings of the work, from which all derived analyses come from, I recommend authors to improve the resolution and size or include a bigger one on the supplementary dataset.

- It would be helpful if Figure 1d legend explicitly cited extended data figure 6 since it provides readers with an extended visualization of genetic and phenotypic correlation results

- Extended Data Figure 9 is cited in line 235 but it does not exist in the supplementary data.

REVIEWER COMMENTS

Reviewer #4 (Remarks to the Author):

The work entitled 'Genetic map of regional sulcal morphology in the human brain' describes the genetic influences on sulcal morphology by using both genome-wide genotyping and whole exome sequencing data on a large UK biobank cohort of more than 40k participants. I commend authors for the solid analytic pipeline designed for the identification of genome wide significant loci (with discovery – replication cohorts), as well as for the permutation procedures to ensure low rate of false positive findings. The selected p-value threshold of 2×10^{-10} accounts for the multiple phenotypes tested and adds credibility to the results. Results are noteworthy, many tools have been used and methodology is solid.

The greatest novelty of the work relies on the discovery and functional characterization of new genetic loci involved in sulcal morphology, which has not been as extensively described as other brain morphology measures such as volume or cortical thinning. Functional and gene expression enrichment analysis across brain tissues, genetic/phenotypic correlations and co-localization analyses give strength and enrich the results. Overall, I think this is potentially a very valuable paper. However, I find the paper sometimes difficult to follow. I have some important concerns related with the genetic characterization of the GWAS results that need to be addressed and some minor comments:

We thank the reviewer for the overall feedback on our manuscript and suggestions for improvement. We also made some minor changes to the text to improve flow.

Major concerns

- I think the weakest point of the paper is the rare variant analysis. What is the advantage provided by using the variants from WES in the association and replication analyses? If there is no clear benefit, I think it would be useful for future studies to state that genotype data can be sufficient to describe the biological underpinnings of similar or related to sulcal phenotypes.
- I think the study lacks a clear rare variants analysis, and I am not very convinced of what is the main contribution of the “coding variant association” section. While using a replication cohort provides a great robustness for the found results (as also mentioned in discussion), it generates a limitation for the detection of rare and private mutations. In the line of the above commented, burden tests for ultra-rare mutations collapsed on specific genes even using the whole cohort could explain some of the gene-based associations with sulcal phenotypes at the rare variation level. Since many neuropsychiatric phenotypes have dense differences between contributions from rare and common genetic variants, it will be interesting to see whether genetic influences on sulcal morphology also differ between rare and common variation. Also, results from rare predisposing variant accumulation tests could illustrate some of the related previous findings cited in the discussion paragraph (line 354 - 363).

We thank the reviewer for these suggestions and comments, and we answer the related points (1) and (2) together below.

In our response to one of the other reviewers, our main focus has been on high-quality, common to low frequency associations to maximize robust findings, with WES included for completeness; we did not anticipate WES to provide many unique associations above and beyond the imputation data at this sample size, hence the limited results. As such, we effectively treated imputed and WES data similarly, as one set, and tried to avoid over-emphasizing the inclusion of WES data which provided marginally added benefits.

From the main single variant results, even the rarest associations mostly have a minor allele frequency (MAF) of ~2% (only 2 bilateral associations have MAF <1%), which is well covered by imputation. The inclusion of single variant level whole exome sequencing data provides only a 2.7% increase in the total number of variants, post-QC (324k out of a total of 11.9 million variants). Additionally, of the 324k WES variants, approximately 227k are also

covered by imputation post-QC, thus the true added variants provided by WES is around 96,000, representing an effective ~0.8% increase in variants (assuming no linkage disequilibrium). Therefore, not surprisingly, all association loci in the main results would have been picked up by imputation. Of course, this could not have been determined *a-priori*, without running the combined analyses as we have done here. Nonetheless, our study is the first to implement WES data into brain imaging traits, and our added results provides better coverage in coding regions for others scientist to perform cross-trait/cross-ancestry fine mapping studies.

The reviewer makes a good point in terms of a gene-based aggregated rare variant analysis, and we have now included a series of burden tests for rare (MAF<1%) loss-of-function protein-truncating associations. This analysis has been added in the **Methods (“Rare variant burden analyses”, page 25) and Results (“Rare variant gene burden associations”, page 9)**, with the addition of **Supplementary Table 9**. Perhaps unsurprisingly, we did not find any significant associations after correcting for the number of sulcal measures ($p < 6.0 \times 10^{-9}$, 0.05/18406/450 sulcal measures), reporting 50 PTV-burden sulcal measure associations at $p < 2.7 \times 10^{-6}$ (0.05/18406 genes tested). Thus, there is little evidence at the current sample size to suggest any significant contribution of rare variant/rare variant burdens to sulcal measures. Therefore, we did not further characterize the impact of rare-variants in the manuscript and retained our focuses on low-frequency to common associations.

We have originally included the following in the discussions:

“We note that the vast majority of observed associations are driven by common variants well-covered by imputation. Therefore, at the current sample sizes, whole-exome sequenced variants provided limited added benefits to imputation for sulcal measures.”,

and now modified with rare-variant burden testing, as follows (**page 15-16, lines 358-364**);

“We note that the vast majority of observed associations were driven by low frequency (MAF>1%) to common variants (MAF>5%) well-covered by genotyping with imputation. Therefore, at the current sample sizes, single rare (MAF<1%) variants and aggregated burden of rare PTVs provide limited additional benefits to imputation for detection of genetic influences on brain sulcal morphology. Potential rare variant associations with large effects may yet to be detected at increased sample sizes.”,

to emphasize the reviewer’s points.

To summarize: The main contribution of the coding variant analysis was to facilitate mapping of significant loci from our genotyping analysis to coding variants, which are likely to be more functional and potential candidates for follow-up. *SLC6A20* is an example of how the integrating of coding variants aids with interpretation; more details are provided in the Supplementary Information.

- When interpreting gene enrichment across neurodevelopmental stages from BrainSpan, only genes downregulated during early infancy are overrepresented. However, instead of focusing on this finding and suggesting reliable explanations, authors focused on some of the genes putatively downregulated during early prenatal stages: DAAM1, NT5C2 and NEO1. These genes, however, seem to reach these highest expression levels at early prenatal stages (color at extended data figure 7 towards brown-red). Also, other genes from the significant list display the opposite pattern in comparison with that observed for the mentioned genes, and this is not described along the manuscript. Therefore, the sentence “These results suggest that genetic effects on regional brain folding are in part driven via regulation of neuronal development during early brain development” is not well demonstrated. This part is confusing and must be revised.

We thank the reviewer for highlighting how this section was confusing. We have reworded the text around the BrainSpan enrichment analyses (**pages 10-11**) to highlight that the significant downregulation is being driven by clusters of genes that are expressed at higher levels pre- versus post-natally, i.e., expressed during the critical period when cortical folding is occurring, rather than only highlighting the downregulation. We have focused on

this result specifically as its expression pattern is the one that is over-represented amongst all of the various expression patterns seen across all genes.

We have also removed the problematic summary sentence (“These results suggest that genetic effects on regional brain folding are in part driven via regulation of neuronal development during early brain development”) from the Results section.

- The gene expression enrichment analyses could even be more refined including some cell-type specific gene-sets, with many recent databases having great information at this respect (see for instance doi.org/10.1093/nar/gkaa76). In the same line, FUMA can also assess for expression enrichment across brain tissues but these results are not reported. Have these results been checked? Are these results similar to those obtained with TissueEnrich R package?

We have now included results for tissue enrichment, which shows down-regulation of the closest gene set across a range of adult tissues including some brain tissues, which is in line with higher expression during early brain development. With regard to cell-type specific analyses, the link provided sent us to the general doi website so we couldn't be sure which database the reviewer was referring too. For internal consistency we performed cell-type specific analyses in FUMA, using a fetal brain cell RNA-seq dataset available with this analysis package. This shows significant enrichment of closest gene set expression in early fetal stem cells, and indicates suggestive enrichment in other cell types at similarly early fetal stages.

All of the FUMA results have now been included in a separate paragraph in the main text (**page 10-11**), and the methods have been updated to reflect the additional analyses. We have also included an additional panel of figures (**Extended Data Figure 8**) for the tissue and cell-type enrichment results. The significant enrichment for brain cortex is seen in both this and the TissueEnrich approach.

- The authors focused on *KCNK2*. Is the multi co-localization described at this locus the norm or a specific case of this phenomenon? Why locus 12q23.3 harboring *NUAK1* gene is not tested when this gene is also highlighted at the results section? In line 144: “Notably, the chr1:215Mb locus (near *KCNK2*) and chr12:106Mb locus (12q23.3, *NUAK1*) were associated with 23 and 22 width measures respectively across multiple brain regions”

The *KCNK2* example mentioned in a section in lines 242-263 (reviewer version) focuses on the example of colocalization with eQTLs visualized in aligned at the regional association level in a hypothesis-driven deeper dive, with the rest of eQTL colocalization results better visualised in the heatmap in **Extended Data Figure 9**. The *NUAK1* locus did not contain a colocalizing eQTL (**Extended Data Figure 9**).

The reviewer makes a good suggestion to test multi-trait colocalization between associations across pleiotropic loci, which we allude to in the main text at line 144-153 (reviewer version). We have incorporated the multi-trait colocalization across all loci associated with ≥ 2 sulcal measures. We found that the majority of multi-sulcal associations at a locus colocalised to the same underlying signal, except for the complex *MAPT* region. Our analyses have been incorporated into the revised manuscript as the new **Supplementary Table 6** and in **Results**, lines 154-159:

“We performed multi-trait colocalization across loci associated with >2 sulcal measures ($n=72$ loci, Supplementary Table 6) and found 62 (86%) of the loci colocalised to a single cluster, 8 (11%) to two clusters, and one to three clusters with the *MAPT-KANSL1* region the outlier where a large number of sulcal associations were not colocalised with others in the region (**Supplementary Table 6**).”

And also in **Discussion**, lines 375-377:

“We also show that the majority of the associations at pleiotropic loci co-localise to one or two shared signals, suggesting largely homogeneous effects at these loci. The *MAPT-KANSL* has an extensive

heterogeneous pleiotropy, in keeping with the complexity at this locus.”

- I miss more discussion related to the genetic correlations between brain sulcal morphology parameters and neuropsychiatric conditions. Although some interesting correlations arise (Parkinson disease or ADHD) and differential patterns across two observable clusters in relation to sulcal width are described, less is said about some intriguing differences. For instance, Schizophrenia and bipolar disorder, with one of the highest SNP-based co-heritability estimates, show opposite correlation with sulcal width. Any explanation for this? Is this a notable difference when comparing with correlation observed with other brain morphology traits? Other notable differences can be seen at Figure 4 between Alzheimer and Parkinson. Also, beyond differential patterns observed, these genetic correlation results could be useful to discuss with neuropsychiatric traits seem to be more impacted by variability at sulcal morphology.

We thank the reviewer for these insightful comments. The original Figure 4a did not annotate the global genetic correlation results with indicators of significance ($p < 0.001$ after adjusting for multiple tests). We have corrected this in the revised version of Figure 4a, adding asterisks indicating statistical significance. The revised figure highlights how the positive global genetic correlation between sulcal width and schizophrenia is significant, but the negative global genetic correlation between sulcal width and bipolar disorder is not significant.

In light of the reviewer’s comments, we have also expanded our discussion of specific genetic correlations (**pages 17-18**), highlighting the differences between Parkinson’s disease (PD) and Alzheimer’s disease (AD) and their opposing genetic correlations with calcarine and occipital lobe surface area and length (positive for PD, negative for AD). We emphasize how similar patterns of positive genetic correlation have previously been observed between PD and total cortical surface area, and hypothesize that this may relate to the radial-unit hypothesis outlined by Rakic et al. (2009):

“The strongest correlations were observed for localised sulcal measures, such as those between anterior inferior temporal sulcus length and ADHD, warranting further investigations in independent datasets. We also observed significant positive genetic correlations between Parkinson’s disease (PD) and various sulcal traits – most notably, with bilateral calcarine and occipital lobe surface area and length – contrasting with significant negative correlations between Alzheimer’s disease and the same sulcal measures. These findings expand upon prior reports of positive genetic correlations between PD and cortical surface area (Grasby et al. (2020)). Our results may indicate divergent genetic contributions to cortical thickness and cortical surface area in PD, whereby increased cortical surface area reflects genetic influences on neural progenitor differentiation, defining the number of neocortical columns, during embryogenesis, whereas reduced cortical thickness reflects events later in development, influencing the number of synapses and neurons per neocortical column (Rakic et al., 2009; Abbasi et al., MedRxiv, 2022).”

Minor comments

- From the beginning of the results section, I understand that the 450 independent sulcal phenotypes selected for GWAS come from the dimensionality reduction with tSNE. The sentence at methods line 457 could be also included in results section to help follow the procedure.

We have added the sentence, “Phenotypes with missingness >75% were excluded from subsequent analysis, leaving 450 measurements (224 left and 225 right hemisphere measures) for analysis”, to the end of the first results section before the GWAS results, as suggested by the reviewer (lines 100-101).

- Figure 1C captures a lot of valuable information but it is very difficult to appreciate in full detail. Since this figure represents one of the main findings of the work, from which all derived analyses come from, I recommend authors to improve the resolution and size or include a bigger one on the supplementary dataset.

Since Figure 1c contains a high-level summary representation of the results, it was difficult to also highlight the detail in the given main figure space. We have included a high-resolution version of this figure in pdf format in a Supplementary File as suggested.

- It would be helpful if Figure 1d legend explicitly cited extended data figure 6 since it provides readers with an extended visualization of genetic and phenotypic correlation results

We have added “Extended phenotypic and genetic correlation heatmap is shown in Extended Data Figure 6” to the end of legends of Figure 1d.

- Extended Data Figure 9 is cited in line 235 but it does not exist in the supplementary data.

We apologise for the typo – this should refer to Extended Data Figure 8 (now the new EDF9), which has now been corrected.

REVIEWER COMMENTS

Reviewer #4 (Remarks to the Author):

I commend authors for having addressed the major and minor concerns I had. I think the paper is now more fluid to read, and the rare-variant analysis performed is now justified with the conclusion stating that rare PTVs provide limited additional benefits to imputation for detection of genetic influences on brain sulcal morphology. I think burden tests performed now strongly helped to clarify this. Although I think the paper is much more improved now, I have however some minor, although important, concerns related to the brain measures and their relation to the genetic results:

- In line 357 , the following sentence: "brain sulcal width has a stronger genetic component and is most stable across the lifespan". Is there any data or previous work to cite that supports this affirmation?

- I am not very sure about what is measured by sulcal width. Is it a proxy for thinner gyri? Do the authors think that sulcal width should be weighted by sulcal length?

- Tsne plotting does, as far as I know, not allow for missing values. Was data interpolated before Tsne? Or imputed? Please

Reviewer comments response

Reviewer #4 (Remarks to the Author):

I commend authors for having addressed the major and minor concerns I had. I think the paper is now more fluid to read, and the rare-variant analysis performed is now justified with the conclusion stating that rare PTVs provide limited additional benefits to imputation for detection of genetic influences on brain sulcal morphology. I think burden tests performed now strongly helped to clarify this.

We thank the reviewer's previous suggestions in strengthening the manuscript and narrative.

Although I think the paper is much more improved now, I have however some minor, although important, concerns related to the brain measures and their relation to the genetic results:

- In line 357 , the following sentence: "brain sulcal width has a stronger genetic component and is most stable across the lifespan". Is there any data or previous work to cite that supports this affirmation?

We have previously demonstrated that sulcal width is the most heritable sulcal measurement, while sulcal length is the least heritable (Pizzagalli et al, Commun Biol, 2020). This and other supporting references have been updated. Based on these heritability estimates, we hypothesized that the pattern of spatial clustering observed in the present study may reflect broad brain topology due to its relatively higher heritability.

Prior studies have also demonstrated how sulcal morphometry is largely determined during prenatal development, and may therefore better reflect genetically influenced, early brain development versus other neuroimaging endpoints. Nonetheless, few studies have explored whether sulcal width is more strongly influenced by genetic factors versus sulcal length or depth. Therefore, we have softened the language in this sentence. The revised sentence includes citations to prior studies of prenatal influences on sulcal measures, in addition to citing the above-mentioned study of sulcal heritability:

"Notably, regional sulcal width measures clustered in a manner that reflected broad brain topology, potentially underlining strong prenatal influences on sulcal development¹⁻⁴, and the relatively higher heritability of sulcal width versus sulcal depth, length, or surface area⁵."

- I am not very sure about what is measured by sulcal width. Is it a proxy for thinner gyri? Do the authors think that sulcal width should be weighted by sulcal length?

We thank the reviewer for highlighting the need for a clearer delineation of the four sulcal shape measures investigated in the present study.

We have added a new paragraph to the Introduction on Page 4, which clearly defines each metric, including sulcal width. We also explain that while measures like sulcal width correlate with cortical thickness, they may be more sensitive to increased age and genetic influences versus cortical thickness and other more commonly used measures of gyral morphometry:

“Using four independent datasets, we recently outlined a range of heritable sulcal measures that can be reliably quantified at high resolution across the whole brain, irrespective of MRI platform or acquisition parameters: Sulcal depth, length, width and surface area. Sulcal depth represents the distance between the cortical surface and the exposed, gyral surface (also known as the hull). Sulcal length represents the distance of the intersection between the medial sulcal surface and the hull. Sulcal width, also known as sulcal span or fold opening, represents the distance between each gyral bank, averaged over all points spanning the median sulcal surface. Sulcal surface area represents a composite of sulcal width, depth and length measures. These sulcal descriptors strongly correlate with measures of thickness in adjacent cortical regions; however, sulcal measures are likely more sensitive to increased age, cognitive performance, and genetic effects compared with more commonly analyzed metrics of gyral morphometry .”

Following on from the above, it is likely that increased sulcal width partly represents a proxy for thinner gyri; indeed, Tang et al. (2021) report that the spatial pattern of cortical thinning in older subjects “partly corresponded with that of sulcal widening”. Nonetheless, sulcal width and other sulcal descriptors likely offer additional insights into cortical morphology beyond measures of gyral thickness alone, as we describe in the new paragraph.

We do not believe that sulcal width should be weighted by sulcal length, as the two measures show differential patterns of development across the lifespan (sulcal shortening is faster before age 40, whereas sulcal widening is faster after age 40) Sulcal surface area, which we have already included in this study, represents a combination of sulcal width, length and depth measures, and likely represents a more appropriate way to weight each sulcal measure by the other(s).

- Tsne plotting does, as far as I know, not allow for missing values. Was data interpolated before Tsne? Or imputed? Please

Prior to TSNE, the missing data was imputed using “imputePCA()” function in R to avoid excessive data loss from differential missingness across all sulcal measures. We have added the clarification to the Methods section (page 22).

References

- 1 Kostovic, I. & Vasung, L. Insights from in vitro fetal magnetic resonance imaging of cerebral development. *Semin Perinatol* **33**, 220-233, doi:10.1053/j.semperi.2009.04.003 (2009).
- 2 Rakic, P. Neuroscience. Genetic control of cortical convolutions. *Science* **303**, 1983-1984, doi:10.1126/science.1096414 (2004).

- 3 Garel, C. *et al.* Fetal cerebral cortex: normal gestational landmarks identified using prenatal MR imaging. *AJNR Am J Neuroradiol* **22**, 184-189 (2001).
- 4 Chi, J. G., Dooling, E. C. & Gilles, F. H. Gyral development of the human brain. *Ann Neurol* **1**, 86-93, doi:10.1002/ana.410010109 (1977).
- 5 Pizzagalli, F. *et al.* The reliability and heritability of cortical folds and their genetic correlations across hemispheres. *Commun Biol* **3**, 510, doi:10.1038/s42003-020-01163-1 (2020).